# The solubility product extends the buffering concept to heterotypic biomolecular condensates

**Aniruddha Chattaraj, Michael L Blinov, Leslie M Loew***

R. D. Berlin Center for Cell Analysis and Modeling, University of Connecticut School of Medicine, Farmington, United States

**Abstract** Biomolecular condensates are formed by liquid-liquid phase separation (LLPS) of multivalent molecules. LLPS from a single ("homotypic") constituent is governed by buffering: above a threshold, free monomer concentration is clamped, with all added molecules entering the condensed phase. However, both experiment and theory demonstrate that buffering fails for the concentration dependence of multicomponent ("heterotypic") LLPS. Using network-free stochastic modeling, we demonstrate that LLPS can be described by the solubility product constant (Ksp): the product of free monomer concentrations, accounting for the ideal stoichiometries governed by the valencies, displays a threshold above which additional monomers are funneled into large clusters; this reduces to simple buffering for homotypic systems. The Ksp regulates the composition of the dilute phase for a wide range of valencies and stoichiometries. The role of Ksp is further supported by coarse-grained spatial particle simulations. Thus, the solubility product offers a general formulation for the concentration dependence of LLPS.

## Introduction

Biomolecular condensates comprise a novel class of intracellular structures formed by a biophysical phenomenon called liquid-liquid phase separation (LLPS) (*Banani et al., 2017*; *Hyman et al., 2014*; *Shin and Brangwynne, 2017*). These structures serve as membraneless compartments where complex biochemistry can be organized and facilitated (*Holehouse and Pappu, 2018*); for example, T cell receptor-mediated actin nucleation efficacy spikes up multifold when all the associated signaling molecules concentrate into a condensate (*Su et al., 2016*) near the plasma membrane. These structures are also implicated in many age-related or neurological diseases (*Shin and Brangwynne, 2017*; *Alberti et al., 2019*; *Mathieu et al., 2020*).

Numerous theoretical and experimental studies have illuminated many of the biophysical requirements for condensate formation (*Li et al., 2012*; *Shin et al., 2017*; *Wang et al., 2018*). In particular, it is firmly established that clustering of weakly interacting multivalent proteins or nucleic acids is a prerequisite for the phase separation. Even a sufficiently concentrated solution of a single self-interacting protein (homotypic interaction) with multiple binding sites in its sequence can partition into protein-dense and dilute phases. Such homotypic systems display a strict threshold concentration above which phase separation occurs. This phase separation serves as a buffering mechanism for the protein in the dilute phase (*Holehouse and Pappu, 2018*; *Klosin et al., 2020*), which remains at the threshold concentration; thus, as more protein is added to the system, the dense phase droplets grow in size and number, keeping the concentration clamped in the dilute phase. Although a homotypic system closely conforms to a single fixed threshold concentration, the picture gets much more complex with multicomponent (heterotypic interactions) systems, which underlie all the biomolecular condensates found in living cells and contain complex mixtures of multivalent proteins and/or nucleic acid. Detailed theoretical analysis of lattice-based simulations explained why the dilute phase

*For correspondence:
les@volt.uchc.edu

Competing interests: The authors declare that no competing interests exist.

concentrations of specific components need not stay fixed when phase separation is driven by heterotypic interactions (*Choi et al., 2019*). This was recently followed by a thorough experimental study of the thermodynamics of the liquid-liquid phase transitions in heterotypic systems, showing clearly that concentration thresholds for phase separation no longer remain fixed and vary with relative compositions of interacting binding partners (*Riback et al., 2020*). But a theoretical framework for quantitatively predicting such complex varying concentration thresholds is still lacking.

We have previously distinguished between strong multivalent interactions, which can produce molecular machines, and weak multivalent interactions, which can produce what we termed 'pleomorphic ensembles' (*Mayer et al., 2009*; *Falkenberg et al., 2013*). The strong binding affinities in molecular machines (e.g., ribosomes, flagella) enforce a specific parts list with a fixed stoichiometry. Pleomorphic ensembles (e.g., cytoskeletal polymers and their associated binding proteins, neuronal post-synaptic densities, etc.) are much more plastic in their composition than molecular machines. Biomolecular condensates are a subclass of pleomorphic ensembles in that their molecular components simultaneously coexist within both a distinct phase and as solutes in the surrounding solution. In trying to quantitatively understand the relationships governing condensate formation from heterotypic components, we asked if there may be lessons to learn from ionic solution chemistry, where salts in solution are in equilibrium with a solid crystalline phase composed of a lattice of counter ions. We realized that the liquid phase separation at threshold concentrations of heterotypic biological molecules might resemble the precipitation of anions and cations from solution.

Consider a salt (let's say silver chloride, AgCl) in water; dissolution takes place until the solution becomes saturated; further addition of salt results in precipitation. In the simplest case of pure AgCl, this seems to be similar to the behavior of a homotypic (single component) condensate; that is, above a threshold total concentration of dissolved AgCl, the salt will not dissolve further, maintaining a clamped concentration of $Ag^+$ and $Cl^-$ in the solution above, whatever amount of AgCl is added. However, the key concept is that the saturation threshold is governed by the solubility product (SP) – the product of the individual concentrations of $Ag^+$ and $Cl^-$, $[Ag^+] * [Cl^-]$. Precipitation starts when this product reaches the thermodynamic parameter called the solubility productconstant, Ksp. For AgCl Ksp = $1.7 \times 10^{-10}$ $M^2$ at 25°C. Importantly, if we add any $Cl^-$ in the form of a highly soluble salt (e.g., KCl) into the solution, some AgCl will precipitate to maintain the solubility product at Ksp. Of course, the crystal lattice in an ionic solid is very different from the set of weak multivalent interactions inside a biomolecular condensate. But we wondered whether the Ksp might at least approximately be used to understand heterotypic LLPS.

We explore how well Ksp may apply to biomolecular condensates using two stochastic modeling approaches. With a non-spatial network-free simulator (NFsim; *Sneddon et al., 2011*), we show how Ksp can approximately predict the phase separation threshold for a two-component system by systematically changing the concentration of individual components with a variety of valencies. We simulate the dynamics of cluster formation, demonstrating a dramatic transition from a stable distribution of small oligomers below Ksp to an unstable bimodal distribution of small oligomers and explosively growing large polymers at or above Ksp. We will also show how a more complex mixed-valent three-component system also conforms to a Ksp. Moreover, these simulation results help explain experiments (*Riback et al., 2020*), where individual components of heterotypic biomolecular condensates are *not* effectively buffered in the dilute phase. We then expand on prior work on the structural features governing multivalent clustering (*Chattaraj et al., 2019*) with a spatial kinetic simulator (SpringSaLaD; *Michalski and Loew, 2016*) to support these conclusions and also point to some limitations.

## Results

### Phase boundary of a two-component system with molecules of the same valency

Our baseline model system consists of a pair of tetravalent molecules ($A_4$ and $B_4$), each having four binding sites that can interact with an affinity (Kd) of 350 µM. We start the NFsim simulation with the same concentrations of monomers of each type and measure the free monomer concentrations when the system reaches the steady state (*Figure 1A*). The product of both free concentrations is called solubility product (SP). As we increase the total concentration (synchronously of both $A_4$ and

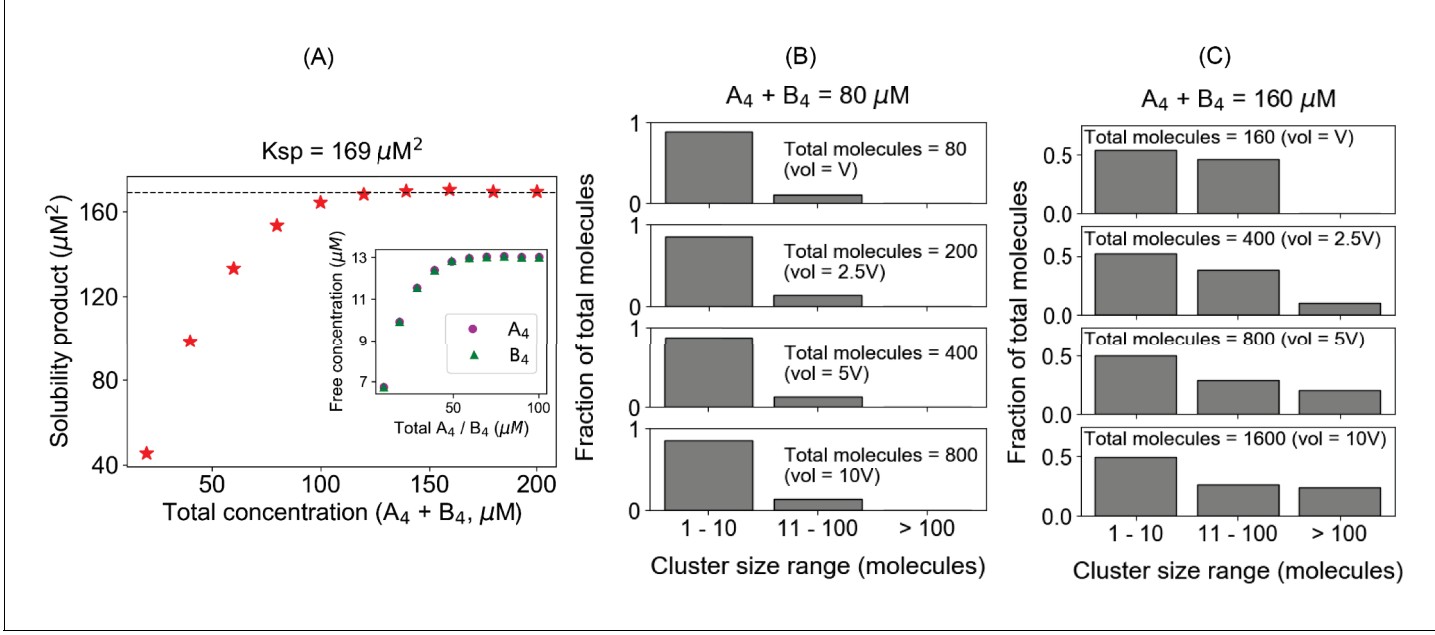

**Figure 1.** The solubility product constant corresponds to a threshold above which molecules distribute into large clusters. These simulation results correspond to equal total concentrations of a heterotypic tetravalent pair of molecules with Kd for individual binding of 350 μM. (A) Product of the free monomer concentrations (solubility product) as a function of the total molecular concentrations. The black dashed line indicates the plateau, corresponding to the solubility product constant (Ksp), 169 μM². Inset plot shows the change of free molecular concentrations of both tetravalent molecules with their respective total concentrations. Each data point is an average of steady-state values from 200 trajectories. In these simulations, we titrate up the molecular counts (200, 400, 600, ...., 2000 molecules, respectively), keeping the system's volume fixed. (B, C) Distribution of cluster sizes with varying system sizes at two different total concentrations, 80 μM and 160 μM, respectively below and above the plateau in (A). The histograms show how the molecules are distributed across different ranges of cluster sizes.

The online version of this article includes the following source data and figure supplement(s) for figure 1:

**Source data 1.** Source data for *Figure 1*.
**Figure supplement 1.** Full (unbinned) distributions of cluster sizes corresponding to *Figure 1B, C*.
**Figure supplement 2.** Pair of tetravalent heterotypic binders shows an identical Ksp even for a larger system.
**Figure supplement 3.** Distributions of cluster sizes for single trajectories.

$B_4$), the concentration of monomeric molecules initially goes up (*Figure 1A*, inset) and of course SP also goes up; upon reaching a concentration threshold (total concentration ~120 μM in this case), SP plateaus to a constant value (169 μM²). This is the solubility product constant or Ksp for this pair of tetravalent molecules.

Next, we ask how the system might behave differently below and above the Ksp. Similar to an analysis we employed in previous work using spatial simulations (*Chattaraj et al., 2019*), we probed for how the number of available molecules (i.e., the system volume) affects the cluster size distribution below and above the threshold, 80 μM and 160 μM (*Figure 1B, C*). If the system shows no tendency to condense into large clusters, the size distribution will be insensitive to the number of molecules at a given total concentration. *Figure 1B* shows this to be the case for the 80 μM total concentration (SP = 154 μM² < Ksp). The shape of the cluster size distribution displays an exponential decline from monomers to higher oligomers, and this shape is insensitive to increasing the number of molecules (i.e., volume) in the system (unbinned histograms are in *Figure 1—figure supplement 1A*). Even in the presence of 800 molecules, there are hardly any clusters greater than 40 molecules (lowest panel of *Figure 1B*). Approximately 80% of the total molecules are in clusters containing less than 10 molecules, no matter how many molecules are available in the system. Extrapolating to a macroscopic system, this would be equivalent to a single soluble phase consisting of mainly monomers and small oligomers.

*Figure 1C* and *Figure 1—figure supplement 1B* illustrate the cluster distribution for total concentration = 160 μM, above the threshold for constant SP (SP = 169 μM² = Ksp). The histogram does change shape, extending the tail to larger clusters eventually to become a bimodal distribution

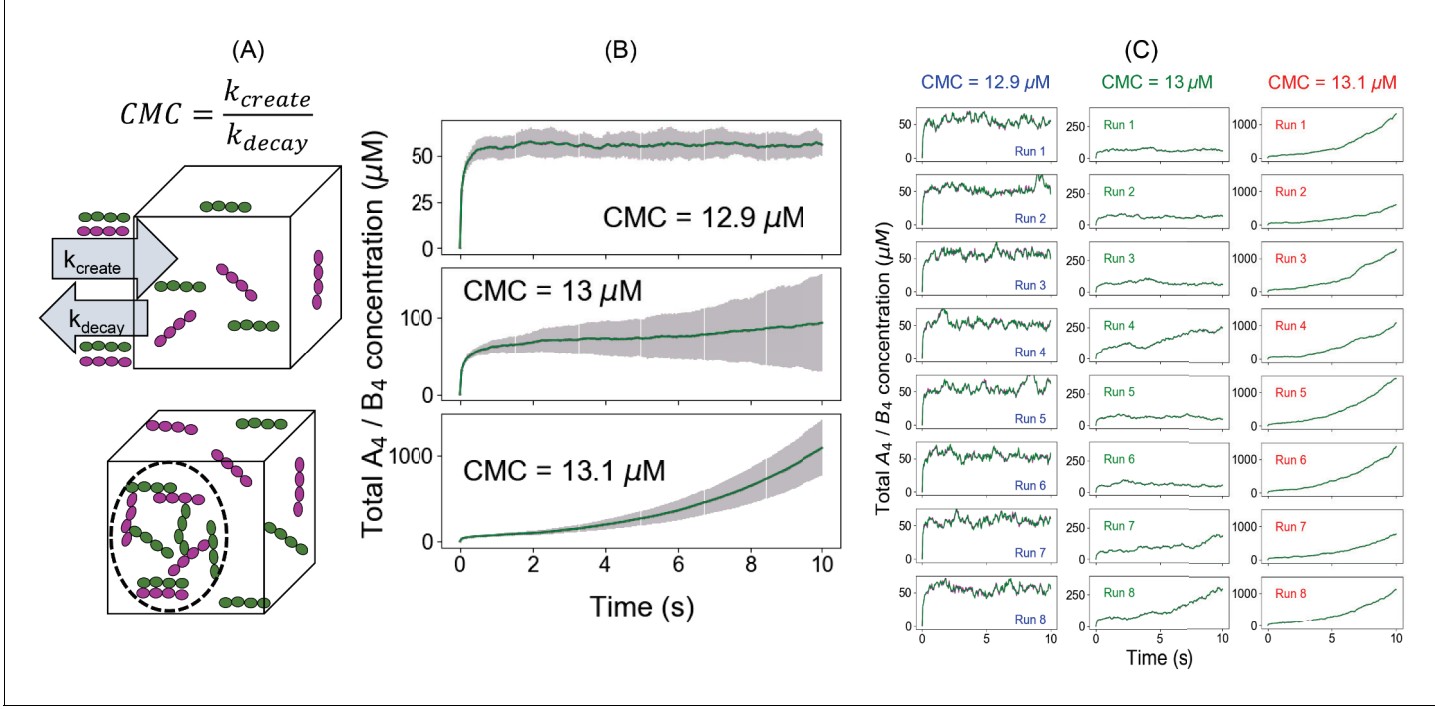

**Figure 2.** An alternative approach to quantify the phase transition boundary. (**A**) Illustration of the clamped monomer concentration (CMC) approach. Both the molecules ($A_4$ in magenta and $B_4$ in green) can enter the simulation box with a rate constant $k_{create}$ (molecules/s) and exit with a rate constant $k_{decay}$ ($s^{-1}$). The ratio of these two parameters clamps the monomer concentration to ($k_{create}/k_{decay}$). (**B**) Average time course (over 100 trajectories) of total molecular concentrations as a function of different CMCs. Error bars show the standard deviations across 100 trajectories. (**C**) Eight sample trajectories for different CMCs.

The online version of this article includes the following source data and figure supplement(s) for figure 2:

**Source data 1.** Source data for *Figure 2*.

**Figure supplement 1.** Individual trajectories of total molecular concentrations ($A_4$ in magenta and $B_4$ in green) at clamped monomer concentration (CMC) = 12.9 µM.

**Figure supplement 2.** Individual trajectories of total molecular concentrations ($A_4$ in magenta and $B_4$ in green) at clamped monomer concentration (CMC) = 13 µM.

**Figure supplement 3.** Individual trajectories of total molecular concentrations ($A_4$ in magenta and $B_4$ in green) at clamped monomer concentration (CMC) = 13.1 µM.

**Figure supplement 4.** Summary of fixed total concentration (FTC) and clamped monomer concentration (CMC) method predictions.

as we feed more molecules into the system (i.e., as we increase the volume). With 1600 molecules in the system, more than 20% of the total molecules are in clusters larger than 100 molecules. *Figure 1—figure supplement 2* shows simulations with 10,000 molecules, averaged over 100 trajectories, systematically varying the total concentrations by changing the simulation volume (as opposed to changing the number of molecules within a fixed volume in *Figure 1A*); the Ksp is still 169 µM² and bimodal distributions clearly develop above Ksp. We note that the long tail in these histograms (unbinned distribution in *Figure 1—figure supplement 1B*) is an average of 100 stochastic trajectories; examination of individual histograms in *Figure 1—figure supplement 3* (note the logarithmic scale on the abscissa) shows almost all of them individually containing just one huge cluster along with small oligomers. Thus, *Figure 1C*, *Figure 1—figure supplement 2B*, and *Figure 1—figure supplement 3* demonstrate that if the system is above Ksp, molecules are funneled into macroclusters.

We hypothesize that this tendency to form increasingly larger clusters with more available molecules is a hallmark of phase separation behavior, as previously established (*Chattaraj et al., 2019*), and that a constant solubility product (Ksp) is a quantitative indicator to mark the threshold that underlies biomolecular condensates. Here, we use the percolation boundary, the threshold for forming large clusters, as a proxy for phase separation, realizing that they may not be completely coincident (*Choi et al., 2020a*; *Choi et al., 2020b*). Below a threshold total concentration, when SP has not reached the constant level of Ksp, the tendency to form large clusters is low and the system

would exist as a single phase (e.g., *Figure 1B*). Above the threshold where SP converges to the Ksp, the system tends to form very large clusters, yielding two different phases, manifest as bimodal cluster size histograms (*Figure 1C*, *Figure 1—figure supplement 2B*, and *Figure 1—figure supplement 3*). The dense phase containing the larger clusters grows in size and the dilute phase concentration remains constant. Importantly, this behavior is precisely that of a buffering system, which has been proposed as one of the important biophysical functions of biomolecular condensates. The generality of this hypothesis will now be further explored through additional modeling scenarios.

## Simulations with monomeric A$_4$ and B$_4$ maintained at fixed concentrations

We now further demonstrate that Ksp marks the phase transition threshold using an alternative modeling approach. In the first approach, we used a fixed total concentration (FTC) of molecules and measure the free monomer concentrations as the system reaches the steady state. In this second approach, we clamp the monomer concentrations to a constant value (clamped monomer concentration ['CMC,], *Figure 2A*) and allow the total concentration (free plus bound) to change over time. This is achieved by creating reactions that rapidly create and destroy monomers, such that the concentration is clamped at the ratio of these rate constants – as long as these rates are much faster than the rates of the binding reactions. The SP, in this case, is simply the product of CMC_A$_4$ and CMC_B$_4$.

For CMC_A$_4$ = CMC_B$_4$ = 12.9 µM (SP = 166.4 µM$^2$, below Ksp), total molecular concentrations rise up initially and then converge to a steady state (~56 µM) (*Figure 2B*). However, going to CMC =

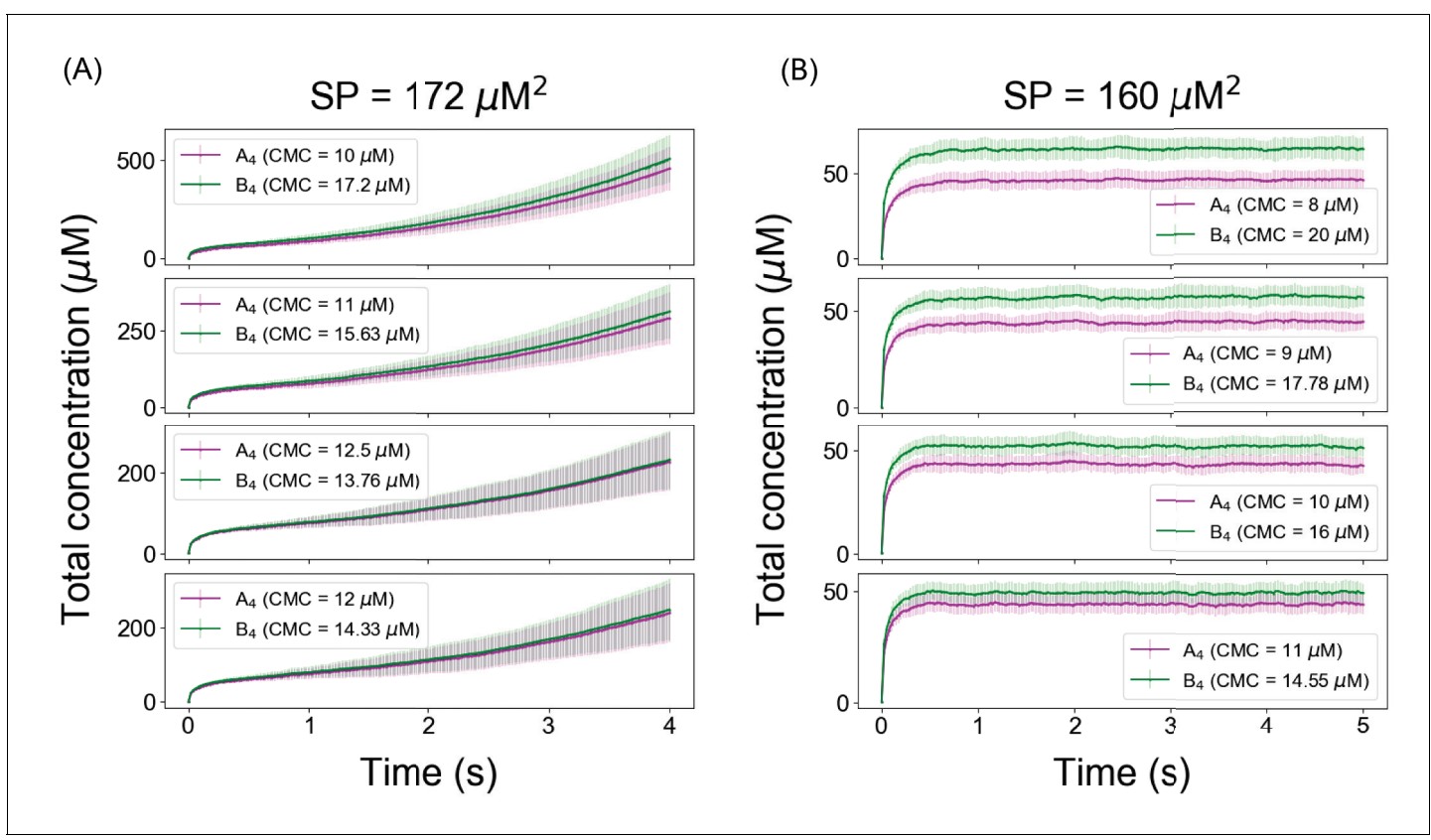

**Figure 3.** The Ksp defines a threshold for unlimited growth of clusters even when the individual concentrations of heterotypic multivalent binding partners are unequal. With Ksp determined from *Figures 1* and *2* at ~169 µM$^2$, solubility product (SP) was clamped above in (A) at 172 µM$^2$ and below in (B) at 160 µM$^2$. The solid lines (magenta and green) and error bars represent the mean and standard deviations across 100 trajectories.

The online version of this article includes the following source data and figure supplement(s) for figure 3:

**Source data 1.** Source data for *Figure 3*.
**Figure supplement 1.** System (A$_4$– B$_4$) deviates from a fixed Ksp when initial conditions are not stoichiometrically matched.

13 µM (SP = 169 µM$^2$ = Ksp), the total concentrations never reach a steady state. This phenomenon is more pronounced for a higher CMC (13.1 µM). Clearly the system is undergoing a fundamental change around SP = 169 µM$^2$, identical to the threshold determined for FTC (*Figure 1*). Both these modeling paradigms indicate that there is a solubility product constant (Ksp = 169 µM$^2$) beyond which the system has a much higher propensity to form larger molecular clusters, the prerequisite for phase-separated droplet formation. Another striking feature is the variability of the total concentrations at CMC = 13 µM as illustrated by the error bars around the mean counts (second panel, *Figure 2B*). When we look at the individual trajectories (*Figure 2C* and *Figure 2—figure supplements 1–3*), the system shows large fluctuations and variable lag times before irreversible growth near the phase boundary (*Figure 2C*, CMC = 13 µM); the total concentration explodes in some runs or fluctuates around a metastable state within the given time frame. The behavior is less stochastic away from the Ksp (*Figure 2B, C*, CMC = 12.9 µM and 13.1 µM) as quantified by the relatively narrower error bars. The behavior at Ksp represents stochastic nucleation of clusters containing enough crosslinking that disassembly becomes unlikely; such larger clusters are sufficiently stable only at or above Ksp.

When we compare the outcomes from FTC and CMC methods, we see that the results are consistent with each other (*Figure 1A*, *Figure 2B,* and *Figure 2—figure supplement 4*). Below the phase boundary, if we clamp the monomer concentrations to the value of free concentrations obtained from the FTC method, we recover the same steady-state total molecular concentrations (top six panels in *Figure 2—figure supplement 4*). However, at even slightly above the Ksp, the CMC total concentration increases with time, rather than leveling off to a higher steady value (bottom panels in *Figure 2—figure supplement 4*).

## Phase transition depends on Ksp even when individual monomer concentrations are unequal

From ionic solution chemistry, we know that irrespective of the individual ionic concentrations, if the product of ion concentrations exceeds the Ksp of the salt (i.e., supersaturation), we always get precipitation to restore the solution concentrations to Ksp, even if one ion is present at a different concentration than the other ('common ion effect'). We wanted to test whether that simple chemical principle works for our relatively complex multivalent molecular clustering system. In *Figure 3*, we analyze two cases when the product of reactant's concentrations (SPs) is above (SP = 172 µM$^2$) and below (160 µM$^2$) the Ksp (169 µM$^2$, derived from *Figure 1*). For each of those SPs, we vary the CMCs of A$_4$ and B$_4$ in such a way that the products of the CMCs are always equal to the assigned SP. Satisfyingly, we find that for SP < Ksp (*Figure 3B*), the systems converge to stable steady states (no phase transition); but for SP > Ksp (*Figure 3A*), irrespective of individual clamped concentrations, systems always show unbounded growth (phase transition). We also titrated unequal FTCs, maintained at a ratio of 3:2 (*Figure 3—figure supplement 1*). Interestingly, in this computational experiment the free monomer concentration of the lower abundant component (B$_4$) gets exhausted disproportionately as the threshold is approached, so that SP cannot quite reach Ksp (169 µM$^2$) and actually begins to diminish somewhat at still higher FTC. However, cluster size distribution (*Figure 3—figure supplement 1B*) still becomes increasingly bimodal with higher concentrations suggesting a phase separating behavior. Thus, even when monomer supply becomes depleted, the Ksp still serves as an upper limit for free monomer concentrations.

## Higher valency promotes phase transition by reducing the Ksp

Increasing valency is known to increase the propensity for phase separation (*Holehouse and Pappu, 2018*; *Mathieu et al., 2020*). Therefore, we ask how the valency of the interacting heterotypic monomers affects the Ksp. We altered the molecular valencies (number of binding sites per molecule) from 3 to 5 and compute the SP profiles as a function of total concentrations (*Figure 4*). The total concentration needed to reach the Ksp goes down with higher valency, consistent with experiment. Going from 3v,3v pair to 4v,4v pair, Ksp changes over fivefold (852 µM$^2$ to 169 µM$^2$), whereas approximately threefold change (169 µM$^2$ to 55 µM$^2$) can be observed for transitioning into 5v,5v pair from 4v,4v.

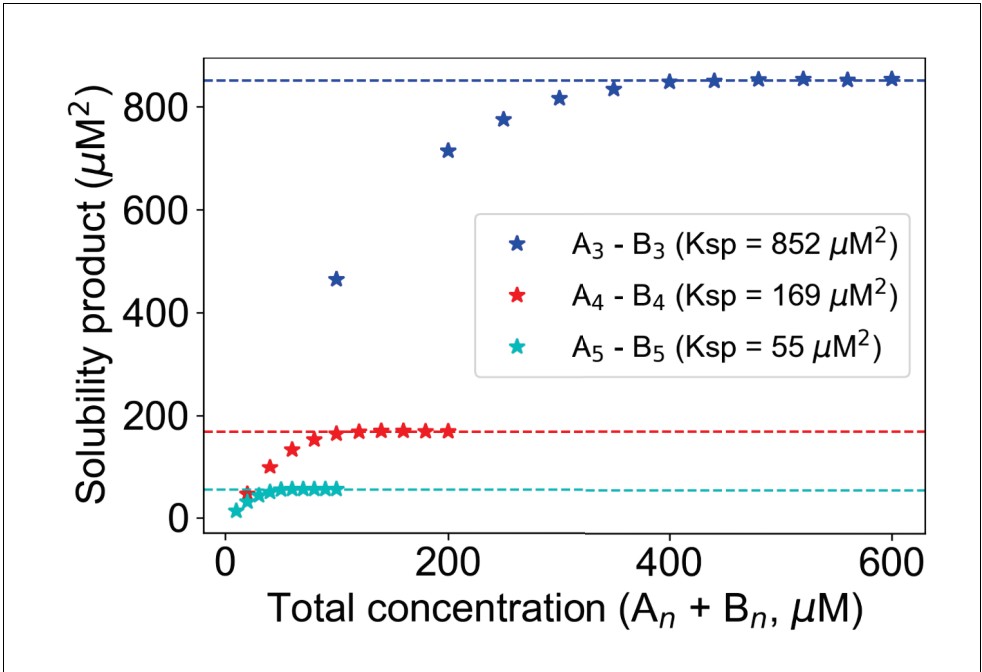

**Figure 4.** Change of Ksp with molecular valency. For 3,3 case (blue stars), molecular counts of each type = [500, 1000, 1500, ..., 3000]. For 4,4 case (red stars), molecular counts of each type = [100, 200, 300, ..., 1000]. For 5,5 case (cyan stars), molecular counts of each type = [50, 100, 150, ..., 500]. Kd is set to 3500 molecules in all these cases. Horizontal dashed lines indicate the Ksp of the corresponding system.

The online version of this article includes the following source data for figure 4:

**Source data 1.** Source data for *Figure 4*.

## Ksp for a mixed-valent system

We next explore what happens when we mix molecules with different valencies. Consider a penta- and trivalent ($A_5$–$B_3$) molecular pair (*Figure 5*). To optimize the clustering such that all sites could potentially be bound would require a stoichiometry of $3A_5$:$5B_3$. Maintaining this concentration ratio, as we titrate up the total concentrations, we see an interesting pattern (*Figure 5A*, inset): the free monomer concentration of $B_3$, which is present in excess, goes up steadily; but the free $A_5$ goes up first and then starts to go down. When we take the product of free monomer concentrations (*Figure 5—figure supplement 1*), we do not see an SP profile that plateaus to a constant Ksp (as in *Figure 1A*). However, when we correct the SP expression by taking the ideal stoichiometry into account, SP = (free $A_5$)$^3$(free $B_3$)$^5$, we get an SP profile that does plateau to a fixed Ksp beyond the total concentration threshold of ~128 µM (*Figure 5A*). The Ksp expression for this mixed-valent binary system is analogous to a mixed-valent salt like $Al_2(SO_4)_3$. Indeed, examining the cluster size distribution confirms that this mixed-valent system has a concentration threshold at the same total concentration 128 µM (48 µM $A_5$ + 80 µM $B_3$) where this stoichiometry-adjusted SP becomes constant; beyond that point the cluster size distribution becomes bimodal and more and more molecules populate the larger clusters (*Figure 5B*). Importantly, the free monomer concentrations do not display buffering above this threshold (*Figure 5A*, inset), with the concentration of the pentavalent monomer actually decreasing. Thus, the Ksp analogy between ionic solution chemistry and biomolecular condensates seems to hold for even these more complex stoichiometries.

## A ternary heterotypic system lacks dilute phase buffering while still being governed by Ksp

In some elegant recent experiments, *Riback et al., 2020* titrated up the concentration of one component of several cellular multicomponent biomolecular condensates and showed that the expected buffering behavior did not pertain to heterotypic systems. We decided to see if our simple non-

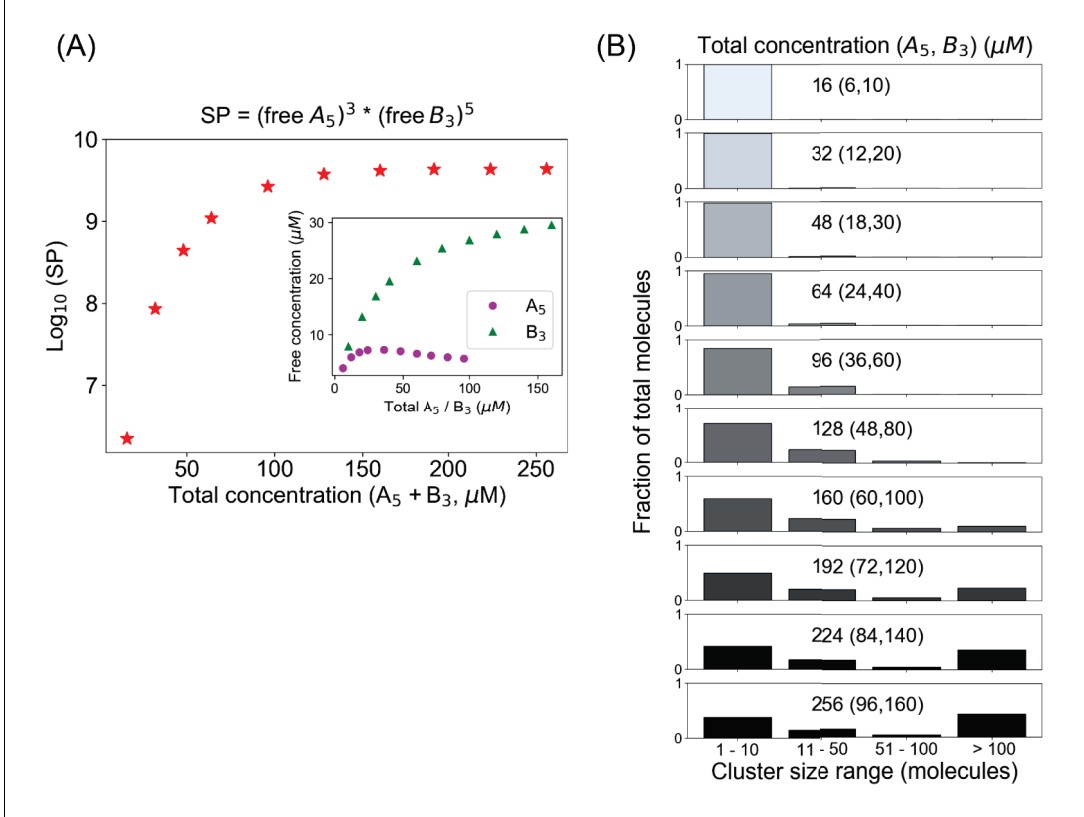

**Figure 5.** A mixed-valent binary system obeys a stoichiometry-adjusted Ksp. (**A**) Logarithm of solubility product (SP) as a function of total concentrations ($A_5 + B_3$). Inset shows the variation of free molecular concentrations w.r.t. their initial total concentrations. Molecules are added at a fixed volume to vary the total concentrations. (**B**) Cluster size distributions become more bimodal as we go beyond the critical concentration (128 µM in this case).

The online version of this article includes the following source data and figure supplement(s) for figure 5:

**Source data 1.** Source data for *Figure 5*.

**Figure supplement 1.** Simple product of free molecular concentrations does not work for mixed-valent binary systems.

spatial simulations, which only consider binding and valency, could recapitulate these experimental observations.

To do this, we consider a mixed-valent three-component system. Component $A_3$ has three sites that can bind to a single domain of component $B_{1,3}$; six sites in component $C_6$ interact with three different sites on component $B_{1,3}$. In our simulations, all these bindings are assumed to have the same weak affinities (Kd = 350 µM). We started by establishing conditions where $B_{1,3}$ and $C_6$ alone could form a bimodal cluster distribution (*Figure 6—figure supplement 1*), corresponding to a phase separation. We found that this binary system has a Ksp of ~2700 µM$^3$ (the units correspond to the ideal stoichiometry of 2 $B_{1,3}$:1 $C_6$). We then chose total concentrations of 120 µM $B_{1,3}$ and 60 µM $C_6$ (well above the phase transition in *Figure 6—figure supplement 1B*) and performed a series of simulations with increasing levels of $A_3$ (*Figure 6*). *Figure 6A–C* shows three ways to analyze these data, which we chose to match the way experimental data for titration of NPM1, a key component of the nucleolus, was analyzed in *Riback et al., 2020* (shown in the corresponding insets in the panels of *Figure 6A–C*). We do not know the valencies and affinities for the components that make up the nucleolus, an archetypal biomolecular condensate, so we made no attempt to match the data quantitatively. However, we are gratified with the obvious qualitative match to the experimental patterns, especially the ability of our simple binding model to show how $A_3$ does not display simple buffering in this scenario. Specifically, buffering, as observed in homotypic biomolecular condensates, would result in a plateau level of monomeric $A_3$ (or NPM1) as a function of total $A_3$ (the illustrative

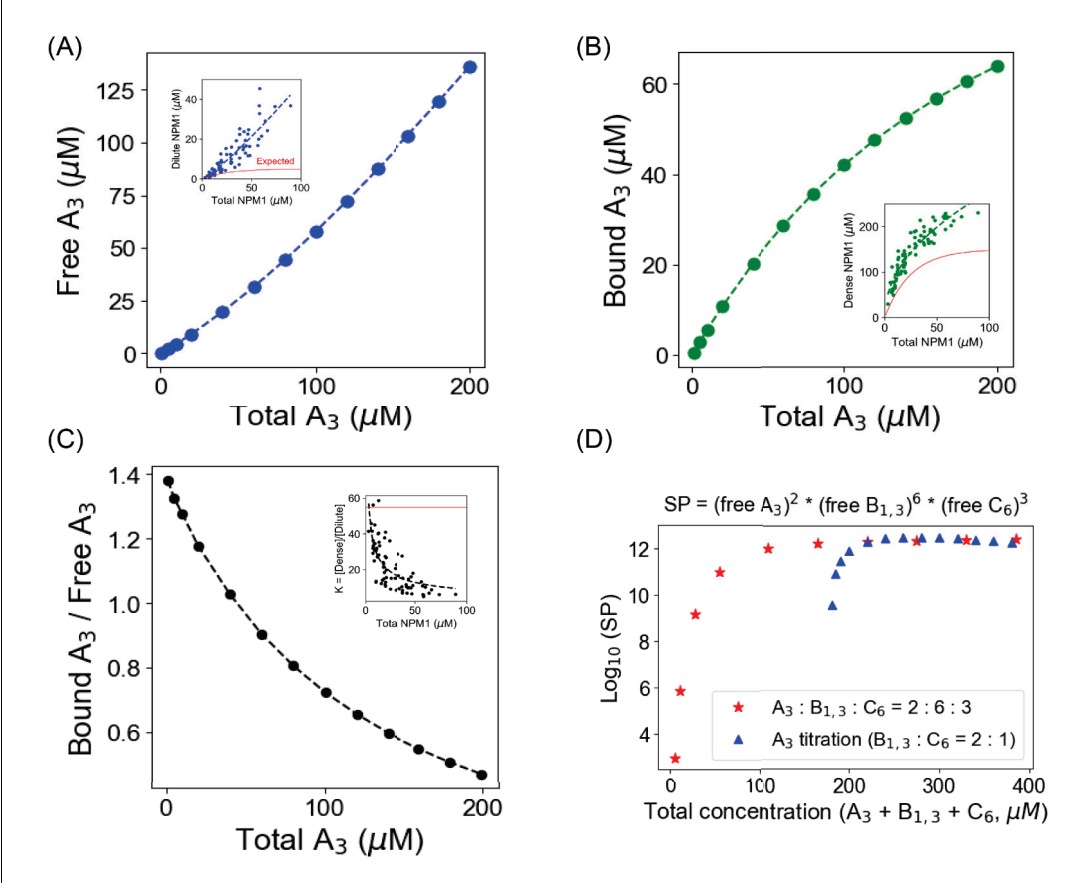

**Figure 6.** Titration of one molecular component in a heterotypic condensate yields correlations with the experimental results [***Riback et al., 2020***]. We begin with 120 μM $B_{1,3}$ and 60 μM $C_6$, which displays a bimodal cluster distribution (condensate formation; Supp. Fig. S6); we then titrate up $A_3$ concentration from 1 μM to 200 μM. (A–C) Free $A_3$ (monomeric) concentration, bound $A_3$ (total $A_3$ – free $A_3$) concentration and their ratio as a function of total $A_3$ concentrations. Inset figures are replotted from the data reported in ***Riback et al., 2020***. To guide the eye, we fit their experimental data to a generic function, $y = a * x^n$ where a and n are pre-exponent and exponent factors, respectively. The red lines in the inset plots demonstrate the 'expected' trend if the condensation is purely driven by homotypic interactions. (D) Blue triangles correspond to the solubility product (SP) of the three-component system when $A_3$ is being titrated up gradually, with fixed total $[B_{1,3}]$ = 120 μM and $[C_6]$ = 60 μM. Red stars indicate the scenario when we simultaneously change concentrations of all three components, keeping a concentration ratio of 2:6:3.

The online version of this article includes the following source data and figure supplement(s) for figure 6:

**Source data 1.** Source data for ***Figure 6***.

**Figure supplement 1.** A pair of trivalent ($B_{1,3}$) and hexavalent ($C_6$) molecules forms condensates when their solubility product reaches to a plateau.

**Figure supplement 2.** Titration profiles for $A_3$ in ternary system.

**Figure supplement 3.** Solubility product(SP) profile for A3–B1,3–C6 system.

'expected' behavior is shown for NPM1 in red in ***Figure 6A***, inset). Instead, our simulations and the corresponding data on the levels of NPM1 in the nucleoplasm (dilute phase) show an increasing concentration of the titrant. Because the dilute phase would contain small oligomers, not just monomer, we confirmed that the patterns in ***Figure 6A, B*** for the monomeric $A_3$ are also present for small oligomers of $A_3$ (***Figure 6—figure supplement 2***).

Importantly, ***Figure 6D*** demonstrates how this complex ternary system obeys the Ksp just as well as the previously analyzed binary systems. The SP for this system is calculated based on the ideal valency matching stoichiometry: SP = $[A_3]^2[B_{1,3}]^6[C_6]^3$. The blue triangles show this analysis for the same simulations that generated ***Figure 6A–C***, where the total concentrations of $B_{1,3}$ and $C_6$ are kept fixed sufficiently high to be phase separated without any $A_3$ (***Figure 6—figure supplement 1B***) at 120 μM and 60 μM, respectively, while the total concentration of $A_3$ is titrated from 1 μM up to 200 μM. The red stars show a computational experiment where the total concentrations of the three

components are varied concertedly, maintaining the ideal stoichiometric ratios of 2 $A_3$:6 $B_{1,3}$:3 $C_6$. Both of these titrations plateau to the same value of Ksp of ~$10^{12}$ $\mu M^{11}$, even though they reach this Ksp at different values of the total concentration. The red star simulations display the characteristic bimodal cluster size distributions at a total concentration of $[A_3] + [B_{1,3}] + [C_6] = (30 + 90 + 45)$ $\mu M$ = 165 $\mu M$, that is, just when SP plateaus to the Ksp (*Figure 6—figure supplement 3*). Thus, even for this more complex mixed-valent ternary system, a threshold for monomer concentrations is set by a Ksp, and it determines the point for formation of the large cluster phase. *Figure 6—figure supplement 3A* also shows how the individual monomer concentrations (insets) continue to change dramatically even after the Ksp is reached. Taken together, these results demonstrate the usefulness of the Ksp concept in explaining the concentrations of individual components of complex heterotypic multivalent binding systems and how they lead to LLPS.

## Spatial simulations

Because of the efficiency of the NFSim non-spatial stochastic simulator, we were able to rapidly explore many scenarios using large numbers of molecules and demonstrate that the solubility product constant (Ksp) may generally serve as a quantitative indicator for phase transitions of multivalent heterotypic binders. These simulations also allowed us to focus on only the effects of binding valency and stoichiometry. We now apply a spatial simulation framework, SpringSaLaD (*Michalski and Loew, 2016*), where the roles of spatial features, such as steric hindrance, molecular flexibility and proximity, may also impact Ksp. A biomolecule is modeled as a collection of spherical sites connected by spring-like linkers. The spheres may be designated as binding sites within a rule-based modeling interface, assigning them macroscopic on and off rates that the software translates to reaction probabilities as the spheres penetrate a computed reaction radius dependent on the on-rate constant. Thus, the software is amenable to computational experiments where the geometric and reaction parameters of the system can be systematically varied.

Our spatial system consists of a pair of matched tetravalent molecules ($A_{4a}$ and $B_{4b}$) where each molecule contains four binding sites, with interspersed pairs of linker sites (*Figure 7A*). These linker sites impart flexibility to the molecules mimicking the intrinsically disordered linker sequences that are found in many phase-separating multivalent proteins (*Posey et al., 2018*). Each of the magenta sites can bind to each of the green sites with an affinity of 350 $\mu M$. We begin with 100 $A_{4a}$ and 100 $B_{4b}$ molecules, randomly distributed in a three-dimensional rectangular volume. As the system relaxes to the steady state, we quantify the monomer concentrations and plot their product (SP) as a function of the initial total concentration – the FTC approach we described above for NFsim. We generate an SP profile by systematically varying the FTC by changing the volume of the compartment, keeping the total molecular numbers same (*Figure 7B*). The SP converges to a constant value (Ksp = 318 $\mu M^2$) at a threshold FTC (~138 $\mu M$ in this case). When we look at the detailed cluster size distributions at steady state (*Figure 7—figure supplement 1A*), the dimer emerges as a preferred configuration both below and above the Ksp. This dimer preference arises from the matching valency and spatial arrangement of the binding sites in the two partner molecules; such an effect, which we term a 'dimer trap,' cannot be realized in non-spatial methods, such as NFsim. Above Ksp, however, most of the individual trajectories, much like NFsim, produce discontinuous distributions with small oligomers along with one or two very larger clusters (*Figure 7—figure supplement 2*). This obvious bifurcation is obscured when averaging over multiple individual histograms (*Figure 7—figure supplement 1A*) because the consistent population of small oligomers is reinforced while individual large clusters will become smeared out.

Having established the Ksp with the FTC spatial simulations, we turned to the CMC approach, again using high values for $k_{create}$ and $k_{decay}$ (as in *Figure 2A*). One can think of the CMC approach as being equivalent to having a large external reservoir of monomers that can rapidly diffuse into a volume where clustering is enabled. As we titrate up the CMCs (*Figure 7C*), the system undergoes a fundamental change at 17.85 $\mu M$, corresponding to SP = 318.6 $\mu M^2$, which is approximately at the Ksp we derived from the FTC calculations. Below that boundary (CMC = 16.7 $\mu M$), total concentrations converge to a steady state (*Figure 7C*, top panel); at (CMC = 17.85 $\mu M$) or above the boundary (CMC = 18 $\mu M$), the total concentration of molecules keeps on going up with time (*Figure 7C*, second and third panels). When we look at the individual total concentration trajectories for CMC = 17.85 $\mu M$ (*Figure 7—figure supplement 3*), much like our NFsim results, some trajectories fluctuate around a lower steady state (dilute phase) for the given time frame while some trajectories shoot up

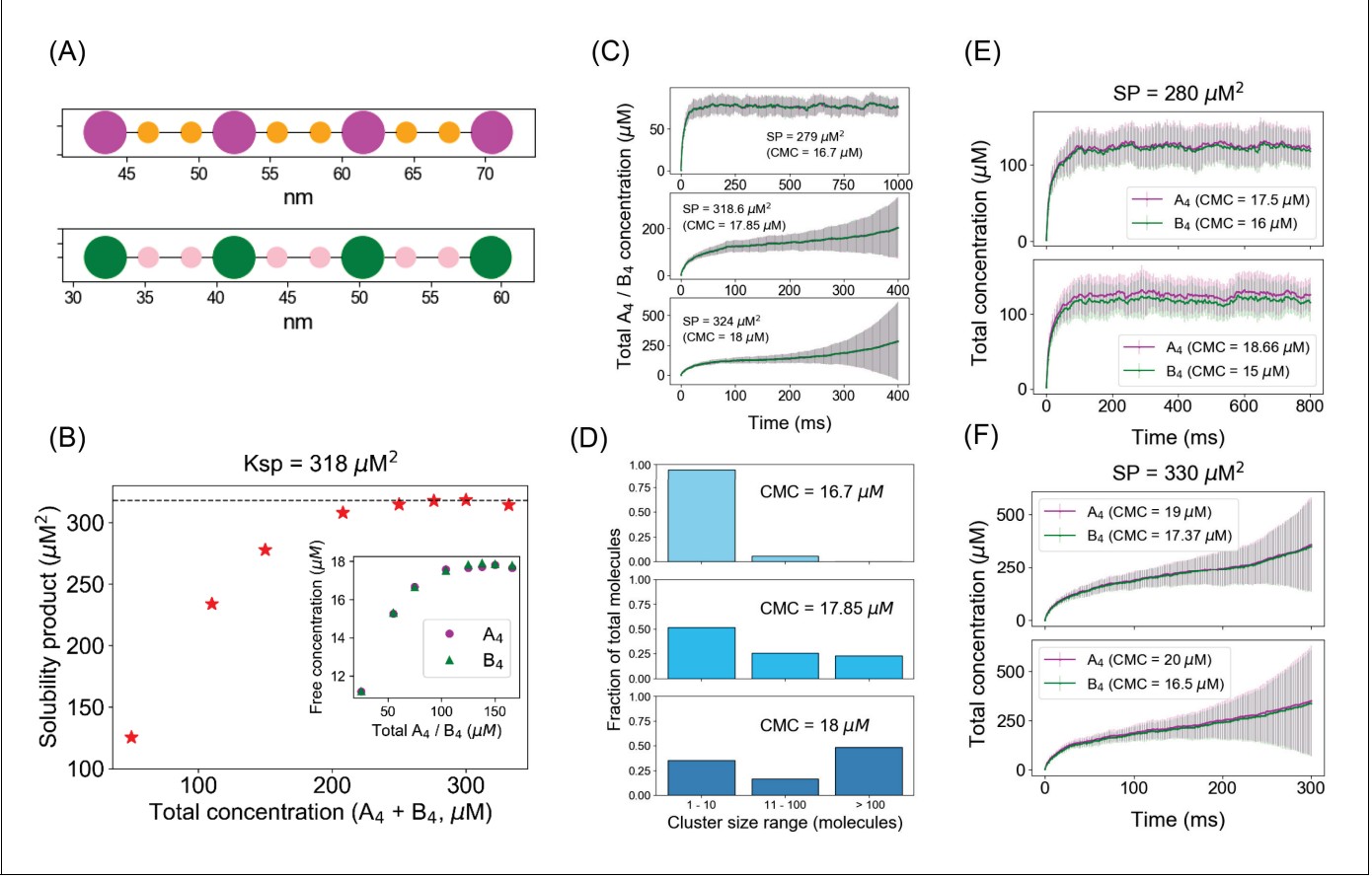

**Figure 7.** Spatial simulations demonstrate similar phase boundary behavior as network-free simulator (NFsim). (**A**) SpringSaLaD representations of a pair of tetravalent binders. $A_{4a}$ and $B_{4b}$ consist of four magenta and green spherical binding sites (radius = 1.5 nm) and six orange and pink linker sites (radius = 0.75 nm). Diffusion constants for all the sites are set to 2 $\mu m^2$/s. For individual binding, dissociation constant, Kd = 350 $\mu$M (Kon = 20 $\mu M^{-1}$. $s^{-1}$, Koff = 7000 $s^{-1}$). Simulation time constants, dt (step size) = $10^{-8}$ $s^{-1}$, dt_spring (spring relaxation constant) = $10^{-9}$ $s^{-1}$. (**B**) Solubility product (SP) profile of the spatial system. We place a total of 200 molecules (100 $A_{4a}$ + 100 $B_{4b}$) in 3D boxes with varying volumes and quantify the monomer concentrations (free $A_{4a}$ and free $B_{4b}$) at steady states. Each data point is an average over 100 trajectories. Solubility product constant (Ksp) = 318 $\mu M^2$, the horizontal dashed line. (**C**) Total molecular concentration profiles for three clamped monomer concentrations (CMCs). The solid lines and error bars represent the mean and standard deviation over 50 trajectories. (**D**) Cluster size distributions at the last time point of CMC trajectories, that is, 1000 ms for CMC = 16.7 $\mu$M, 400 ms for CMC = 17.85 $\mu$M and 18 $\mu$M. More detailed histograms without binning are shown in **Figure 7—figure supplement 1**. (**E, F**) Irrespective of individual monomer concentrations, total molecular concentrations converge to steady states as long as the solubility product (SP = 280 $\mu M^2$) < Ksp (**E**) and diverge with time when SP (= 330 $\mu M^2$) > Ksp (**F**). The solid lines and error bars represent the mean and standard deviation over 50 trajectories.

The online version of this article includes the following video, source data, and figure supplement(s) for figure 7:

**Source data 1.** Source data for **Figure 7**.

**Figure supplement 1.** For SpringSaLaD system, (**A**) cluster size distributions at steady state (sampled from last time point) for different fixed total concentrations; (**B**) cluster size distributions at last time point for different clamped monomer concentrations.

**Figure supplement 2.** Cluster size distributions for single SpringSaLaD trajectories.

**Figure supplement 3.** For SpringSaLaD system, individual total concentration trajectories at (**A**) clamped monomer concentration (CMC) = 16.7 $\mu$M (below Ksp), (**B**) CMC = 17.85 $\mu$M (at Ksp), and (**C**) CMC = 18 $\mu$M (above Ksp).

**Figure 7—video 1.** Single trajectory for spatial simulations below Ksp.

https://elifesciences.org/articles/67176#fig7video1

**Figure 7—video 2.** Single trajectory for spatial simulations at Ksp.

https://elifesciences.org/articles/67176#fig7video2

**Figure 7—video 3.** Single trajectory for spatial simulations slightly above Ksp.

https://elifesciences.org/articles/67176#fig7video3

**Figure 7—video 4.** Dynamics of individual clusters below (left panel) and above (right panel) the Ksp.

https://elifesciences.org/articles/67176#fig7video4

(dense phase) after a variable lag. This stochastic behavior is also present for CMC = 18 µM, although the rate of growth is much faster in this case. As quantified by the error bars (*Figure 7C*, second and third panels), stochastic fluctuations are somewhat greater in spatial simulations than the non-spatial scenario. This stochasticity can be attributed to the variable lag before the nucleation of a sufficiently large cluster for irreversible and accelerating growth; higher variability in spatial simulations is expected due to a larger number of contributing factors like steric crowding and optimal geometric conformations of binding sites.

Certain features of these results are better appreciated via *Figure 7—videos 1–3*, each corresponding to a typical single trajectory at the three CMCs. Interactive 3D visualizations of these three simulations are available on the 'Simularium' website hosted by the Allen Institute for Cell Science; readers may access them here: Below Ksp; At Ksp; Above Ksp. The videos each show the actual dynamics of cluster formation and diffusion within the 3D volume, synchronized with the time course of total molecular concentration and a dynamic histogram of cluster size distribution. *Figure 7—video 1*, where monomer concentration is clamped at 16.7 µM (SP = 279 µM$^2$, below Ksp), shows that the total molecular concentration fluctuates around ~80 µM, matching the third data point in *Figure 7B*. While this single trajectory is noisy, it corresponds well to the average of 50 trajectories in the top panel of *Figure 7C*. Importantly, the dynamic histogram in *Figure 7—video 1* shows that the system rarely samples a cluster size greater than 15 molecules, which we associate with a single dilute phase. *Figure 7—video 2* displays a typical trajectory with CMC at the Ksp (monomer concentration 17.85 µM, SP = Ksp = 318.6 µM$^2$). Instead of the steady state attained below Ksp (*Figure 7—video 1*), *Figure 7—video 2* displays a noisy but accelerating increase in total concentration to a maximum of 500 µM at 400 ms and a corresponding filling of the simulation volume; the dynamic histogram (note the logarithmic scale of the x-axis compared to *Figure 7—video 1*) shows primarily small clusters until about 200 ms, followed by a steady siphoning of newly appearing monomers into a single large cluster, which we associate with phase separation. *Figure 7—video 3*, with CMC set just above Ksp at 18 µM, illustrates how one trajectory reaches a metastable steady state that lasts until ~180 ms, but ultimately explodes to almost 800 µM total concentration, virtually filling the available volume (as also noted for the corresponding averaged trajectories in the lowest panel of *Figure 7C*). This corresponds to a nucleation step, which lasts until the formation of a sufficiently large cluster to capture most newly appearing monomers. Motions and spatial locations of individual clusters can be visualized through *Figure 7—video 4*, which is based on the same simulations used to produce *Figure 7—video 1* and *Figure 7—video 3* (i.e., below and above Ksp, respectively). *Figure 7—video 4* displays the individual clusters in a given time frame by computing their centroids and a radius of gyration around that center. For visual clarity, the cluster sizes are scaled down proportionately (by a factor of 4); for example, the largest cluster in the last time frame of the above Ksp (right panel) has a radius of gyration of ~48 nm. The dimension of the simulation volume is 100 * 100 * 120 nm$^3$. Through *Figure 7—video 4*, we can better appreciate the dramatically different spatial distributions of clusters below and above Ksp. The left panel, below Ksp, corresponds to a collection of small clusters homogeneously distributed across the simulation volume (single dilute phase), while the right panel illustrates the evolution of a large cluster (dense phase) coexisting with a pool of randomly distributed small clusters (dilute phase).

Like our non-spatial simulations, both FTC and CMC approaches yield self-consistent results for the spatial system: as the monomer SP remains below the Ksp threshold (318 µM$^2$), the system exhibits only one phase; but above that threshold boundary, the molecules get partitioned into two different phases – a dilute phase with monomers and small oligomers and a highly clustered phase (*Figure 7D* and *Figure 7—figure supplement 1B*). The validity of the Ksp is preserved even when the CMCs are unequal. We illustrate this by choosing two SPs, 280 µM$^2$ and 330 µM$^2$, respectively below and above the Ksp, and varying the individual CMCs (*Figure 7E, F*). Both the cases with SP = 280 µM$^2$ converge to a steady state, while SP = 330 µM$^2$ combinations explode in both cases. It should be noted that as the total concentration explodes in these simulations, the volume can become filled with newly created molecules; this puts a brake on the total concentration in long duration simulations.

## Discussion

It has been widely understood that for single component, self-interacting (homotypic) multivalent systems undergoing liquid-liquid phase separation (LLPS), the concentration of monomer in the dilute phase remains fixed above the phase transition, no matter how much of the monomer is added to the system; this feature of biomolecular condensates underlies buffering and noise reduction (*Holehouse and Pappu, 2018*; *Klosin et al., 2020*). Of course, the maintenance of a fixed concentration in a saturated solution of a solute in equilibrium with its solid phase is an elementary thermodynamic rule. Recognizing this, we set out to see how far the analogy to solution chemistry could take us in considering heterotypic interactions between different multivalent binders. Somewhat more complex than the saturated solution of a single solute is the precipitation of solid salts from saturated solutions of their ions. Chemists well know that ionic solutions of weakly soluble salts are governed by the solubility product constant, $Ksp = [C^{m+}]^n[A^{n-}]^m$, where m and n are the valencies, respectively, of the cation $C^{m+}$ and anion $A^{n-}$; importantly, n and m are the stoichiometries, respectively, of the cation and the anion in the solid phase to balance the positive to the negative charges. The solubility product constant derives from a fundamental thermodynamic principle – equality of chemical potential between coexisting phases (ions in solution and solid). We wondered whether similar expressions could define the thresholds for LLPS in multicomponent (heterotypic) multivalent interaction systems.

One limitation in the analogy is that the composition of the ionic solid phase, and therefore the activity, is invariant, making the system-free energy dependent on only the activities of the ions in solution. However, the ideal stoichiometry, which absolutely constrains the composition of an ionic solid, is not so strictly enforced in the condensed phase of a multivalent condensate because of the weak binding affinities that underlie these systems. Therefore, to explore how well the Ksp might apply to LLPS, we used a non-spatial stochastic network fee simulator, NFsim (*Sneddon et al., 2011*); it isolates only the effect of valency and binding on the concentration dependence of clustering. We used a single weak binding affinity (Kd = 350 μM) for all our simulations. The efficiency of this computational method also made it possible to screen many scenarios with a sufficient number of molecules and trajectories to assure statistically that we were not missing any interesting effects. We used two approaches to assess the threshold behavior. First, we titrated up the total concentration of pairs of multivalent binders. We found that there was a threshold above which the concentration of free monomers obeyed a Ksp expression – that is, $[A]^n[B]^m$, where m and n are the ideal stoichiometries for an oligomer with all binding sites occupied. Below the Ksp threshold, the histogram of cluster size distributions tails off exponentially from monomers to small oligomers and its shape is independent of the number of molecules available (e.g., *Figure 1B* and *Figure 1—figure supplement 1A*). However, for total concentrations above the Ksp threshold, the histogram of cluster size distributions becomes bimodal, with an increasing population of huge clusters as more molecules are added to the system (e.g., *Figure 1C* and *Figure 1—figure supplement 1B*); we consider this bimodal cluster size distribution to be a hallmark of phase separation. The individual trajectories show a separation between small oligomers and a single large cluster (*Figure 1—figure supplement 3*). This bifurcation between one large cluster and small oligomers has been used to define a percolation boundary, generally considered a proxy for phase separation (*Choi et al., 2019*; *Choi et al., 2020a*). To further relate the behavior of these stochastic systems to the macroscopic phase transition, we used a second modeling approach where we clamped the monomer concentrations (CMC) while allowing the clusters to grow. When monomer concentrations were clamped below the Ksp, the system reached a steady state identical to that of the corresponding closed fixed total concentration simulations (*Figure 2—figure supplement 4*), with identical cluster distributions containing a single decaying histogram of cluster sizes. However, when the CMC was set to the Ksp or slightly above it, the plot of total concentration vs. time exploded following an initial lag (e.g., *Figure 2*), indicating that as new monomers enter the system they were funneled into large clusters without attaining a steady state. Together, we feel that the behavior of these two different modeling approaches shows that the Ksp defines a threshold of monomer concentrations above which a phase separation occurs.

The generality of the Ksp as an indicator of threshold was then further tested using different scenarios. We tested a situation where the total concentrations of each molecule in the binary tetravalent heterotypic system were unequal; while these simulations showed that the Ksp determined in

the analysis of the equal concentration system was obeyed for a range of unequal concentrations (*Figure 3*), there will be deviations if the range is extended too far (*Figure 3—figure supplement 1*). This deviation is not surprising considering that the binding affinities are weak, leaving an excess of binding sites empty within a cluster, when the free concentration of one binding partner is too highly depleted. We tested cases where the valencies were not identical for a binary (*Figure 5*) and even a ternary system of mixed valency interactors (*Figure 6D*). In these systems, a more complex Ksp formulation was required to account for the appropriate ideal stoichiometry of the cluster. In both of these cases, the Ksp was successful in defining a threshold for phase separation.

The ternary system of *Figure 6* also allowed us to explore how well the simple multivalent binding simulations could reproduce some recent experiments demonstrating how the components of heterotypic biomolecular condensates are *not* effectively buffered in the dilute phase (*Riback et al., 2020*). For several prototypical cellular LLPS systems, this study elegantly showed that titration of a single component produced a free concentration of that component that increased even more rapidly than its total concentration (*Figure 6A* inset is an example). The simulation results in *Figure 6* show that our simple multivalent binding system is able to reproduce these experimental titration patterns; importantly, it also further demonstrates, despite this failure of simple buffering, that the concentrations of individual components in the dilute phase are still constrained by Ksp.

Recent computational (*Choi et al., 2019*) and theoretical (*Deviri and Safran, 2021*) studies demonstrated that buffering of dilute phase concentrations in multicomponent systems has a complex relationship with the interplay of homotypic and heterotypic interactions. For a two-component heterotypic system, plotting the total concentrations of one component against the other produces a phase diagram with an elliptical region corresponding to the coexistence of the dilute and condensed phases; the system has a single phase anywhere outside that ellipse. In fact, the dilute and condensed phase concentrations remain constant (i.e., buffered) along tie lines that are approximately parallel to the major axis of the ellipse; buffering fails perpendicular to tie lines (*Deviri and Safran, 2021*). The derivation of approximate order parameters, such as a percolation boundary (*Choi et al., 2019*; *Choi et al., 2020b*), to estimate the shape of phase diagrams, could be possible with our approach, but it is beyond the scope of this work. However, the SP should be approximately constant (Ksp) within the two-phase elliptic region. That is, the SP may be used to predict concentrations in the dilute phase even for short traversals *perpendicular* to tie lines in the phase diagram.

We turned to coarse-grained spatial simulations of clustering to determine if the Ksp might still be generally applicable when the shape and flexibility of multivalent molecules is explicitly considered. We used SpringSaLaD (*Michalski and Loew, 2016*) software; it models molecules as a series of linked spheres to represent domains within macromolecules. We had previously used this software to address the structural features that control clustering of multivalent molecules and showed that above a concentration threshold, the system display unlimited growth characteristic of LLPS (*Chattaraj et al., 2019*). In the present study, we examined a heterotypic pair of tetravalent binders (*Figure 7*), similar to the non-spatial model used in *Figures 1* and *2*. The SpringSaLaD structures included four binding spheres, each separated by a pair of linker spheres; the distances between sites were identical within each binding partner. The results (*Figure 7*) show that this model displays the same behavior noted for the non-spatial model. The product of concentrations of monomer becomes independent of the total concentration above a threshold (*Figure 7B*). When monomer concentrations are clamped at or above this Ksp threshold, the total concentration explodes after a lag time (*Figure 7C, F*) and the histogram of cluster sizes becomes bimodal (*Figure 7E*). These results are dramatically displayed in *Figure 7—videos 1–3*, corresponding to typical single trajectories for the clamped monomer concentrations below, at, and above Ksp, respectively; *Figure 7—video 4* offers a view directly comparing cluster sizes as a function of time below above Ksp. Thus, the Ksp concept remains valid for the spatial system shown in *Figure 7A*.

Manipulation of computational models thus allowed us to systematically determine how well the Ksp concept, familiar from ionic solution chemistry, might apply to the very different situation of weak interactions between multivalent macromolecules. We found that for most scenarios, if not all, the Ksp does define a threshold for the unbounded growth of large clusters and a quantitative metric for the tendency of a system to phase separate. Additionally, for those cases where there are deviations from the stereotypical behavior, insights may be gleaned into the underlying molecular factors inhibiting cluster growth. But it is important to appreciate that experimental systems may

introduce additional levels of complexity, such as nonspecific low-affinity binding interactions and long-range electrostatic forces. For real biomolecular condensates, the valency and stoichiometry of the components may be unknown, and they may have multiple binding interaction of differing affinities. However, it should be possible to measure Ksp experimentally using fluorescently labeled binding partners and determining their concentrations in the dilute and condensed phase via quantitative confocal microscopy (*Fink et al., 1998*). Titration experiments can readily be performed in vitro and have been among the earliest studies of biomolecular condensates (*Li et al., 2012*). More recently, methods have been developed to perform titrations in cells (*Riback et al., 2020*). Such experiments could serve, initially, to validate the Ksp concept. Beyond validation, enough such data may make it possible to find optimal fits to a Ksp expression, providing an indication of the effective valency of the interactions in complex multivalent biomolecular condensates.

## Materials and methods

### Non-spatial simulations (NFsim)

To develop models that probe for the effects of valency and concentrations but do not account for spatial effects, we employ the NFsim (*Sneddon et al., 2011*) – a non-spatial rule-based stochastic simulation framework where each biomolecule represents a molecular object that may have multiple binding sites. These sites can bind with other sites depending on a set of rules defined in the model. The simulations have units of molecular counts, but these can be readily converted to equivalent concentrations, which is how we present our results.

The NFsim model file is specified in BioNetGen Language (BNGL; http://bionetgen.org/). Let us take the example of tetravalent binders – $A_4$ (a1, a2, a3, a4) and $B_4$ (b1, b2, b3, b4). We need 16 binding rules to define all the bimolecular interactions, each having an affinity (Kd) of 350 µM. Now 1000 molecules of $A_4$ and $B_4$ with an affinity of 3500 molecules would translate to 100 µM molecular concentrations with 350 µM binding affinity. There are two equivalent ways to change the molecular concentrations: (1) change the Kd, keeping the molecular counts same, which is mathematically equivalent to changing the volume of the system; (2) change the molecular counts, keeping the Kd same. We utilize both approaches for our simulations and specify which is used in our descriptions of the results. We chose binding rules to only allow inter-molecular binding; we felt this was appropriate because NFsim cannot account for spatial proximity of binding sites or steric crowding within clusters. Once the BNGL file is defined, we then generate the corresponding XML file, which serves as the NFsim input file. We run multiple stochastic simulations in parallel using the high-performance computing facility at UConn Health (https://health.uconn.edu/high-performance-computing/). A single NFsim run (500 ms, FTC approach), containing 1000 $A_4$ and $B_4$ molecules each, took ~1 min with 100 trajectories run in parallel. A Python script is then used to perform statistical analysis across all the trajectories.

### Spatial simulations (SpringSaLaD)

To account for realistic spatial geometry, we employ SringSaLaD (*Michalski and Loew, 2016*) – a particle-based spatial simulation platform where each biomolecule is modeled as a collection of hard spheres connected by stiff spring-like linkers. The simulation algorithms are fully described (*Michalski and Loew, 2016*), and we previously studied various spatial biophysical factors in the context of multivalent biomolecular cluster formation (*Chattaraj et al., 2019*) with this software. SpringSaLaD also uses a rule-based method to define binding reactions between multivalent binders.

The SpringSaLaD model files are generated using the graphical user interface (GUI) of the software (https://vcell.org/ssalad). We define the size of binding sites, distance between the binding sites and the overall shape of the molecule inside the GUI. To build a spatial version of the reference system ($A_{4a}$ and $B_{4b}$), two linear tetravalent molecules are constructed first, each having four binding sites and six linker sites. Unlike NFsim, in SpringSaLaD, one binding rule between 'A_type' and 'B_type' sites can take care of all the possible binding interactions. Also, we can define the binding affinity in concentration units (350 µM) directly inside the GUI. We initialize our system in a 3D rectangular geometry; for example, 100 molecules in a $10^6$ nm$^3$ cubic volume (X = Y = Z = 100 nm) would correspond to 166 µM. We change the volume of our system to alter the molecular concentrations. Once the model is specified, as before, we run multiple stochastic simulations in parallel using

our high-performance computing facility. Execution time is very much sensitive to the number of total sites due to the computational overhead of tracking individual site locations. A typical run (50 ms, FTC approach), containing 100 molecules each of $A_{4a}$ and $B_{4b}$ (total sites = 2000), took ~6 hr.

### Data analysis and visualization

Python scripts are used to analyze and visualize the data. All the scripts are written with Spyder IDE (version 4.0.0) (https://www.spyder-ide.org/). Frequently used Python libraries are numpy 1.17.3, pandas 0.25.3, and matplotlib 3.1.2. All the packages are managed by anaconda package distributions (https://www.anaconda.com/).

All the model files, Python scripts, and a 'Readme' description of all the contents are available in a public GitHub repository: https://github.com/achattaraj/Ksp_phase_separation, (copy archived at swh:1:rev:22643ca2ed21b527ccdedbe6a99c2cfc29780df8), *Chattaraj, 2021* .

## Acknowledgements

We thank our colleague Boris Slepchenko for thoughtful discussions on the clamped monomer concentration approach. We thank Blair Lyons and Graham Johnson of the Allen Institute for Cell Science for developing and hosting the Simularium interactive visualizations of our SpringSaLaD simulations. We gratefully acknowledge support of this work by NIH grants R01 GM132859 and R24 GM137787 from the National Institute for General Medical Sciences.

## Additional information

### Funding

| Funder | Grant reference number | Author |
| --- | --- | --- |
| National Institute of General Medical Sciences | R24 GM137787 | Leslie M Loew |
| National Institute of General Medical Sciences | R01 GM132859 | Leslie M Loew |

The funders had no role in study design, data collection and interpretation, or the decision to submit the work for publication.

### Author contributions

Aniruddha Chattaraj, Conceptualization, Data curation, Software, Formal analysis, Investigation, Visualization, Methodology, Writing - original draft; Michael L Blinov, Resources, Investigation, Methodology, Writing - review and editing; Leslie M Loew, Conceptualization, Resources, Supervision, Funding acquisition, Validation, Investigation, Writing - original draft, Project administration, Writing - review and editing

### Author ORCIDs

Aniruddha Chattaraj (iD) https://orcid.org/0000-0002-7105-6621
Michael L Blinov (iD) https://orcid.org/0000-0002-9363-9705
Leslie M Loew (iD) https://orcid.org/0000-0002-1851-4646

### Decision letter and Author response

Decision letter https://doi.org/10.7554/eLife.67176.sa1
Author response https://doi.org/10.7554/eLife.67176.sa2

## Additional files

### Supplementary files

• Transparent reporting form

## Data availability

All the model files, Python scripts and a "Readme" description of all the contents are available in a public GitHub repository: https://github.com/achattaraj/Ksp_phase_separation (copy archived at https://archive.softwareheritage.org/swh:1:rev:22643ca2ed21b527ccdedbe6a99c2cfc29780df8). Also source data files are given for 7 figures that are part of the manuscript.

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
