## [Decision Letter]

**Acceptance summary:**

The concept of solubility products will likely be useful as a tool for experimentalists studying the prospect of buffering via condensate formation in systems driven by heterotypic interactions.

**Decision letter after peer review:**

Thank you for submitting your article "Solubility product governs the concentration threshold for formation of biomolecular condensates" for consideration by *eLife*. Your article has been reviewed by 3 peer reviewers, one of whom is a member of our Board of Reviewing Editors, and the evaluation has been overseen by José Faraldo-Gómez as the Senior Editor. All reviewers have opted to remain anonymous.

Essential Revisions (for the authors):

1. Please refocus the title, abstract and scope to highlight the proposal/finding that the solubility product "rescues" the concept of buffering even in systems where heterotypic interactions drive phase transitions. The consensus is that the solubility product might be rather useful, if it can be readily calculated, to set expectations regarding thresholding/buffering in systems with heterotypic interactions, i.e., all biomolecular condensates.

2. Please improve the overall scholarship to place the work in the appropriate context. Please note that Choi et al., previously explained why the dilute phase concentrations of specific components need not stay fixed when phase separation is driven by heterotypic interactions. Please see Figure 12 and the accompanying discussion in https://journals.plos.org/ploscompbiol/article?id=10.1371/journal.pcbi.1007028.

Please also note that Deviri and Safran have introduced a physical theory for how buffering can be achieved when phase separation is driven by heterotypic interactions. Please see, please cite, and please include a suitable discussion of:

https://www.biorxiv.org/content/10.1101/2021.01.05.425486v1.

3. Please eliminate all remarks about freshman chemistry or high school chemistry.

4. Please provide a mechanistic explanation for the implications of solubility products setting thresholds.

5. The consensus is that the section on the effects of linkers detracts from the focus of the manuscript. A detailed and rigorous treatment of this subject has been published in *eLife* by Harmon et al.,. We recommend that the revised manuscript exclude all results beyond Figure 7. This, we believe, will sharpen the focus, highlight novel aspects, and avoid overlap with work that has come before.

6. The reviewers have raised specific concerns and weaknesses regarding the solubility product. It is a useful concept that touches base with percolation transitions, but not necessarily with the rigor of concepts underlying phase transitions. Specifically, the connections between solubility products and slopes of tie lines needs to be established. We believe that this is beyond the scope of the current manuscript, but it will be very helpful to discuss this issue in a revised Discussion section. Please also note that formal order parameters are not being used in either simulation paradigm for defining phase separation.

7. Please provide a synthesis to explain how either simulation paradigm and the calculation of solubility products can be used to analyze experimental data. In doing so, please spell out the type of experimental data that will come in handy in testing the hypothesis that the solubility product sets an upper limit in systems with heterotypic interactions, thereby enabling the field to assess where the buffering capacity lies in distinct multicomponent systems.

8. And please consider presenting analyses in terms of amplitudes of fluctuations. These will likely help in paring down the number of panels that one presents for each system.

*Reviewer #1 (Recommendations for the authors):*There, unfortunately, are far too many concerns to be raised with regard to the technical and theoretical aspects of this work. A lot of new jargon terms are introduced without connecting to established terminology. The work on linkers is a minimalist redux of that of Harmon et al. The "dimer trap", also explained by Harmon et al., is a redux of the work of Wingreen and colleagues. The unique insight here has to do with the finding / proposal that the solubility product sets an upper limit on the joint concentrations, and hence appears to rescue the concept of buffering in systems with heterotypic interactions. Recent work from Deviri and Safran and from at least one other lab have introduced the concept of heterotypic buffering and provided rigorous thermodynamic descriptions, in the language and formalism of slopes of tie lines, focusing on chemical and mechanical equilibria to explain the full range of buffering capacities realizable as a consequence of the interplay between homotypic and heterotypic interactions. If we remove all the parts of the current manuscript that are best described as redux, then we are left with the solubility product and the rescue of buffering as the novel insight. What is lacking is a rigorous connection to the concepts of phase separation and percolation, achievable only through the calculation of full phase diagrams or at least coexisting curves and the demonstration that the solubility product is invariant along tie lines, and it should be true for all tie lines.

*Reviewer #2 (Recommendations for the authors):*

– The resolution of the cluster sizes is quite low for the non-spatial simulations, showing only 4 bins, whereas it is much higher for the spatial ones. Why so low?

– The movies are helpful in observing the dynamics, but what are the Spring constants?

– Does Ksp still hold in Figure 5 when the Stoichiometry of components is not 5:3?

– In Figure 6, what is the fit function to the dots?

– Do all sites retain excluded volume, even when they are bound to other sites?

– What rate constants were used? What was the integration time-step? Are the results sensitive to these choices?

– I think mentioning freshman/high school chemistry once in the paper is enough. The repeated references in the discussion to the age at which we all should have learned this start to sound like a failure of the field. A text book reference could instead emphasize that the concept is old and well established.

*Reviewer #3 (Recommendations for the authors):*The authors claim that the solubility product is a simple yet useful concept to study the phase boundary, but in fact, the solubility product is a specific realization of a much more general concept of the reaction quotient (Q) and the equilibrium constant (K), which is also from freshman chemistry. Hence, their simulation data just show that at low concentrations Q is smaller than K, and after Q reaches K, it remains constant. What is the benefit of using Ksp instead of more general K?

Unfortunately, it seems to me that there are almost no new findings relevant to the field of phase separation biology. Importance of the linker flexibility and the sticker spacing in phase separation have been known for a while, and the valency effect as well. However, the technical advancement (or applications) might be useful to the computational field, so I strongly recommend this paper for publication in a more technical journal for computational work. I don't think that this manuscript meets the *eLife* criteria.

---

## [Author Response]

Essential Revisions (for the authors):1. Please refocus the title, abstract and scope to highlight the proposal/finding that the solubility product "rescues" the concept of buffering even in systems where heterotypic interactions drive phase transitions. The consensus is that the solubility product might be rather useful, if it can be readily calculated, to set expectations regarding thresholding/buffering in systems with heterotypic interactions, i.e., all biomolecular condensates.

We appreciate this advice to fully focus our paper on the idea that the Ksp can serve as a more general concept than buffering. We have eliminated the additional results (Figure 8) on how the Ksp might be influenced by structural features of the interacting molecules. Of course, the Ksp reduces to simple buffering for homotypic systems, so the last comment of the editor that our idea applies to all biomolecular condensates is also helpful. We believe we have captured this refocus in our new title and abstract:

“The solubility product extends the buffering concept to heterotypic biomolecular condensates.”

“Biomolecular condensates are formed by liquid-liquid phase separation (LLPS) of multivalent molecules. LLPS from a single (”homotypic”) constituent is governed by buffering: above a threshold, free monomer concentration is clamped, with all added molecules entering the condensed phase. However, both experiment and theory demonstrate that buffering fails for the concentration dependence of multi-component (“heterotypic”) LLPS. Using network-free stochastic modeling, we demonstrate that LLPS can be described by the solubility product constant (Ksp): the product of free monomer concentrations, accounting for the ideal stoichiometries governed by the valencies, displays a threshold above which additional monomers are funneled into large clusters; this reduces to simple buffering for homotypic systems. The Ksp regulates the composition of the dilute phase for a wide range of valencies and stoichiometries. The role of Ksp is further supported by coarse-grained spatial particle simulations. Thus, the solubility product offers a general formulation for the concentration dependence of LLPS.”

We have extensively revised the manuscript to focus on this message, to address the other issues raised by the reviewers and to further clarify the presentation. The major changes that address the Reviewers’ comments are highlighted in red in the revised manuscript.

2. Please improve the overall scholarship to place the work in the appropriate context. Please note that Choi et al., previously explained why the dilute phase concentrations of specific components need not stay fixed when phase separation is driven by heterotypic interactions. Please see Figure 12 and the accompanying discussion in https://journals.plos.org/ploscompbiol/article?id=10.1371/journal.pcbi.1007028.Please also note that Deviri and Safran have introduced a physical theory for how buffering can be achieved when phase separation is driven by heterotypic interactions. Please see, please cite, and please include a suitable discussion of:https://www.biorxiv.org/content/10.1101/2021.01.05.425486v1.

We became aware of the very relevant work in Choi et al. as a result of Reviewer 1’s comment (thank you) and now cite it in 3 places. We were aware of the Devri and Safran preprint, which was submitted after our original preprint submission. We have updated our paper with a citation and discussion of this work (see response to item 6).

3. Please eliminate all remarks about freshman chemistry or high school chemistry.

As chemists (2 of the authors), we find it exciting that this elementary chemistry principle could be extended to help explain the complex concentration dependence of LLPS; however, we now explain this without referring to “freshman” chemistry.

4. Please provide a mechanistic explanation for the implications of solubility products setting thresholds.

We have added additional explanations within the discussion, which also further highlight some of the limitations of the Ksp idea:

“The of solubility product constant derives from a fundamental thermodynamic principle – equality of chemical potential between coexisting phases (ions in solution and solid). One limitation in the analogy is that the composition of the ionic solid phase, and therefore the activity, is invariant, making the system free energy dependent on only the activities of the ions in solution”

See also the response to item 6.

5. The consensus is that the section on the effects of linkers detracts from the focus of the manuscript. A detailed and rigorous treatment of this subject has been published in eLife by Harmon et al.,. We recommend that the revised manuscript exclude all results beyond Figure 7. This, we believe, will sharpen the focus, highlight novel aspects, and avoid overlap with work that has come before.

We have complied and believe it has sharpened the focus.

6. The reviewers have raised specific concerns and weaknesses regarding the solubility product. It is a useful concept that touches base with percolation transitions, but not necessarily with the rigor of concepts underlying phase transitions. Specifically, the connections between solubility products and slopes of tie lines needs to be established. We believe that this is beyond the scope of the current manuscript, but it will be very helpful to discuss this issue in a revised Discussion section. Please also note that formal order parameters are not being used in either simulation paradigm for defining phase separation.

We have added the following paragraph to the Discussion (referencing both Choi and Devri):

“Recent computational [13] and theoretical [23] studies demonstrated that buffering of dilute phase concentrations in multicomponent systems has a complex relationship with the interplay of homotypic and heterotypic interactions. For a two-component heterotypic system, plotting the total concentrations of one component against the other produces a phase diagram with an elliptical region corresponding to the coexistence of the dilute and condensed phases; the system has a single phase anywhere outside that ellipse. In fact, the dilute and condensed phase concentrations remain constant (i.e. buffered) along tie-lines that are approximately parallel to the major axis of the ellipse; buffering fails perpendicular to tie lines [23]. The derivation of approximate order parameters, such as a percolation boundary [13, 21], to estimate the shape of phase diagrams, could be possible with our approach, but it is beyond the scope of this work. However, the solubility product should be approximately constant (Ksp) within the two-phase elliptic region. That is, the solubility product may be used to predict concentrations in the dilute phase even for short traversals perpendicular to tie lines in the phase diagram.”

7. Please provide a synthesis to explain how either simulation paradigm and the calculation of solubility products can be used to analyze experimental data. In doing so, please spell out the type of experimental data that will come in handy in testing the hypothesis that the solubility product sets an upper limit in systems with heterotypic interactions, thereby enabling the field to assess where the buffering capacity lies in distinct multicomponent systems.

We have expanded the concluding sentences to suggest how the Ksp concept could lead to new experimental insights:

“However, it should be possible to measure Ksp experimentally using fluorescently labelled binding partners and determining their concentrations in the dilute and condensed phase via quantitative confocal microscopy [24]. Titration experiments can readily be performed in vitro and have been among the earliest studies of biomolecular condensates [9]. More recently, methods have been developed to perform titrations in cells [14]. Such experiments could serve, initially, to validate the Ksp concept. Beyond validation, enough such data may make it possible to find optimal fits to a Ksp expression, providing an indication of the effective valency of the interactions in complex multivalent biomolecular condensates.”

8. And please consider presenting analyses in terms of amplitudes of fluctuations. These will likely help in paring down the number of panels that one presents for each system.

Done. Individual trajectories displaying varying time lags for nucleation, are now available in Supporting Figures. Because of this interesting variable nucleation time, we felt it was important to include them.

Reviewer #1 (Recommendations for the authors):There, unfortunately, are far too many concerns to be raised with regard to the technical and theoretical aspects of this work. A lot of new jargon terms are introduced without connecting to established terminology. The work on linkers is a minimalist redux of that of Harmon et al. The "dimer trap", also explained by Harmon et al., is a redux of the work of Wingreen and colleagues. The unique insight here has to do with the finding / proposal that the solubility product sets an upper limit on the joint concentrations, and hence appears to rescue the concept of buffering in systems with heterotypic interactions. Recent work from Deviri and Safran and from at least one other lab have introduced the concept of heterotypic buffering and provided rigorous thermodynamic descriptions, in the language and formalism of slopes of tie lines, focusing on chemical and mechanical equilibria to explain the full range of buffering capacities realizable as a consequence of the interplay between homotypic and heterotypic interactions. If we remove all the parts of the current manuscript that are best described as redux, then we are left with the solubility product and the rescue of buffering as the novel insight. What is lacking is a rigorous connection to the concepts of phase separation and percolation, achievable only through the calculation of full phase diagrams or at least coexisting curves and the demonstration that the solubility product is invariant along tie lines, and it should be true for all tie lines.

As described above, we have refocused the paper on how the solubility product offers a more general framework than buffering for constraining the concentrations of multivalent molecules undergoing LLPS. We also attempted to better place our work, which is more chemistry oriented, in the context of previous work on the physics of LLPS.

Reviewer #2 (Recommendations for the authors):– The resolution of the cluster sizes is quite low for the non-spatial simulations, showing only 4 bins, whereas it is much higher for the spatial ones. Why so low?

Corrected. Importantly, we now also show histograms for single FTC simulations at steady state (Figure 1 —figure supplement 3); these show clear bifurcations between an exponentially decreasing distribution at low cluster sizes in equilibrium with a single cluster at a very high molecular count.

– The movies are helpful in observing the dynamics, but what are the Spring constants?

We use the default value throughout: dt_spring = 10-9 s^-1^ (see input file in the Git repository). The algorithm only uses the stiff springs to transmit the Langevin forces between linked spheres; their actual deformation is negligible. See [19] for details.

– Does Ksp still hold in Figure 5 when the Stoichiometry of components is not 5:3?

Figure 3 —figure supplement 1 shows that when we deviate from ideal stoichiometry (for A4 – B4, it is 1:1), SP reaches a maximal level close to the Ksp derived from the ideal stoichiometry. The maximal point does represent the onset of phase transition. We did not try this for a titration with a non-ideal stoichiometry for the system in Figure 5, but expect it to behave similarly.

– In Figure 6, what is the fit function to the dots?

The data is fitted to a generic function y = a*x n where a and n are pre-exponent and exponent factors respectively. This was done just to guide the eye, as in the experimental data from Riback, et al.

– Do all sites retain excluded volume, even when they are bound to other sites?

In SpringSaLaD, all the sites retain excluded volumes whether bound or unbound. In other words, at each simulation time step, the solver checks for any physical overlap, irrespective of the binding status of the sites.

– What rate constants were used? What was the integration time-step? Are the results sensitive to these choices?

All the parameters can be found in simulation input files, available in a public github directory. https://github.com/achattaraj/Ksp_phase_separation.

Kon = 20 μM-1.s^-1^, K_off_ = 7000 s^-1^, dt = 10-8 s^-1^

The time courses are of course sensitive to these choices, but the paper focuses on comparisons in steady state behavior and these are sensitive to only Koff/Kon. Of course all these values were kept the same for all simulations.

– I think mentioning freshman/high school chemistry once in the paper is enough. The repeated references in the discussion to the age at which we all should have learned this start to sound like a failure of the field. A text book reference could instead emphasize that the concept is old and well established.

We have deleted all these comments.

Reviewer #3 (Recommendations for the authors):The authors claim that the solubility product is a simple yet useful concept to study the phase boundary, but in fact, the solubility product is a specific realization of a much more general concept of the reaction quotient (Q) and the equilibrium constant (K), which is also from freshman chemistry. Hence, their simulation data just show that at low concentrations Q is smaller than K, and after Q reaches K, it remains constant. What is the benefit of using Ksp instead of more general K?

All of the points in our FTC titration diagrams are at equilibrium. The reaction quotient (Q) refers to a system that is out of equilibrium, allowing the chemist to predict how concentrations will change as equilibrium is approached, so we do not understand the relevance of this comment. Furthermore, we don’t see how an equilibrium constant, K for these complex systems could be formulated. Because of multivalency, we are dealing with systems containing multiple cascading reversible reactions and an infinite number of possible species and also with a system that undergoes a phase transition (or percolation transition). The key point in our treatment is that the SP, which can be determined empirically, converges to an approximate constant at the concentration threshold for LLPS (or more precisely, the percolation threshold). We are sorry that Reviewer 3 missed this key point and hope that the revised manuscript will help her/him understand the novelty of our study.

Unfortunately, it seems to me that there are almost no new findings relevant to the field of phase separation biology. Importance of the linker flexibility and the sticker spacing in phase separation have been known for a while, and the valency effect as well. However, the technical advancement (or applications) might be useful to the computational field, so I strongly recommend this paper for publication in a more technical journal for computational work. I don't think that this manuscript meets the eLife criteria.

Reference

1. Banani, S.F., et al., Biomolecular condensates: organizers of cellular biochemistry. Nature Reviews Molecular Cell Biology, 2017. 18: p. 285.

2. Hyman, A.A., C.A. Weber, and F. Jülicher, Liquid-Liquid Phase Separation in Biology. Annual Review of Cell and Developmental Biology, 2014. 30(1): p. 39-58.

3. Shin, Y. and C.P. Brangwynne, Liquid phase condensation in cell physiology and disease. Science, 2017. 357(6357).

4. Holehouse, A.S. and R.V. Pappu, Functional Implications of Intracellular Phase Transitions. Biochemistry, 2018. 57(17): p. 2415-2423.

5. Su, X., et al., Phase separation of signaling molecules promotes T cell receptor signal transduction. Science (New York, N.Y.), 2016. 352(6285): p. 595-599.

6. Shin, Y. and C.P. Brangwynne, Liquid phase condensation in cell physiology and disease. Science, 2017. 357(6357): p. eaaf4382.

7. Alberti, S., A. Gladfelter, and T. Mittag, Considerations and Challenges in Studying Liquid-Liquid Phase Separation and Biomolecular Condensates. Cell, 2019. 176(3): p. 419-434.

8. Mathieu, C., R.V. Pappu, and J.P. Taylor, Beyond aggregation: Pathological phase transitions in neurodegenerative disease. Science, 2020. 370(6512): p. 56.

9. Li, P., et al., Phase transitions in the assembly of multivalent signalling proteins. Nature, 2012. 483: p. 336.

10. Shin, Y., et al., Spatiotemporal Control of Intracellular Phase Transitions Using Light-Activated optoDroplets. Cell, 2017. 168(1): p. 159-171.e14.

11. Wang, J., et al., A Molecular Grammar Governing the Driving Forces for Phase Separation of Prion-like RNA Binding Proteins. Cell, 2018. 174(3): p. 688-699.e16.

12. Klosin, A., et al., Phase separation provides a mechanism to reduce noise in cells. Science, 2020. 367(6476): p. 464.

13. Choi, J.-M., F. Dar, and R.V. Pappu, LASSI: A lattice model for simulating phase transitions of multivalent proteins. PLOS Computational Biology, 2019. 15(10): p. e1007028.

14. Riback, J.A., et al., Composition-dependent thermodynamics of intracellular phase separation. Nature, 2020. 581(7807): p. 209-214.

15. Mayer, B.J., M.L. Blinov, and L.M. Loew, Molecular machines or pleiomorphic ensembles: signaling complexes revisited. J Biol, 2009. 8(9): p. 81.1-81.8.

16. Falkenberg, Cibele V., Michael L. Blinov, and Leslie M. Loew, Pleomorphic Ensembles: Formation of Large Clusters Composed of Weakly Interacting Multivalent Molecules. Biophys J, 2013. 105(11): p. 2451-2460.

17. Sneddon, M.W., J.R. Faeder, and T. Emonet, Efficient modeling, simulation and coarse-graining of biological complexity with NFsim. Nature Methods, 2011. 8(2): p. 177-183.

18. Chattaraj, A., M. Youngstrom, and L.M. Loew, The Interplay of Structural and Cellular Biophysics Controls Clustering of Multivalent Molecules. Biophys J, 2019. 116(3): p. 560-572.

19. Michalski, P.J. and L.M. Loew, SpringSaLaD: A Spatial, Particle-Based Biochemical Simulation Platform with Excluded Volume. Biophysical journal, 2016. 110(3): p. 523-529.

20. Choi, J.-M., A.S. Holehouse, and R.V. Pappu, Physical Principles Underlying the Complex Biology of Intracellular Phase Transitions. Annual Review of Biophysics, 2020. 49(1): p. 107-133.

21. Choi, J.M., A.A. Hyman, and R.V. Pappu, Generalized models for bond percolation transitions of associative polymers. Phys Rev E, 2020. 102(4-1): p. 042403.

22. Posey, A.E., A.S. Holehouse, and R.V. Pappu, Chapter One – Phase Separation of Intrinsically Disordered Proteins, in Methods in Enzymology, E. Rhoades, Editor. 2018, Academic Press. p. 1-30.

23. Deviri, D. and S.A. Safran, Physical theory of biological noise buffering by multi-component phase separation. bioRxiv, 2021: p. 2021.01.05.425486.

24. Fink, C., F. Morgan, and L.M. Loew, Intracellular fluorescent probe concentrations by confocal microscopy. Biophys J, 1998. 75(4): p. 1648-58.